# Neutrophil Extracellular Traps in Cardiovascular and Aortic Disease: A Narrative Review on Molecular Mechanisms and Therapeutic Targeting

**DOI:** 10.3390/ijms25073983

**Published:** 2024-04-03

**Authors:** Nahla Ibrahim, Wolf Eilenberg, Christoph Neumayer, Christine Brostjan

**Affiliations:** Division of Vascular Surgery, Department of General Surgery, Medical University of Vienna, University Hospital Vienna, 1090 Vienna, Austria; nahla.ibrahim@meduniwien.ac.at (N.I.); wolf.eilenberg@meduniwien.ac.at (W.E.); christoph.neumayer@meduniwien.ac.at (C.N.)

**Keywords:** abdominal aortic aneurysm, aortic disease, cardiovascular disease, inflammation, neutrophil extracellular traps (NETs)

## Abstract

Neutrophil extracellular traps (NETs), composed of DNA, histones, and antimicrobial proteins, are released by neutrophils in response to pathogens but are also recognized for their involvement in a range of pathological processes, including autoimmune diseases, cancer, and cardiovascular diseases. This review explores the intricate roles of NETs in different cardiovascular conditions such as thrombosis, atherosclerosis, myocardial infarction, COVID-19, and particularly in the pathogenesis of abdominal aortic aneurysms. We elucidate the mechanisms underlying NET formation and function, provide a foundational understanding of their biological significance, and highlight the contribution of NETs to inflammation, thrombosis, and tissue remodeling in vascular disease. Therapeutic strategies for preventing NET release are compared with approaches targeting components of formed NETs in cardiovascular disease. Current limitations and potential avenues for clinical translation of anti-NET treatments are discussed.

## 1. Introduction

Neutrophils, the most abundant type of leukocytes in humans, play a crucial role in the innate immune response, acting as the first line of defense against microbial pathogens and participating in the inflammatory reaction [1]. In response to various stimuli such as pathogen-associated patterns, inflammatory mediators, and oxidative stress, neutrophils release neutrophil extracellular traps (NETs), which are web-like structures composed of DNA, histones, and other cellular proteins [2]. Initially described as a mechanism to ensnare and kill microbes, NET formation has since been recognized for its involvement in numerous pathological processes beyond infection, including thrombosis, autoimmune diseases, cancer, and cardiovascular diseases (CVDs) [3].

The interplay between neutrophils, NETs, and CVDs has garnered significant attention in more recent years, also in understanding the pathogenesis of abdominal aortic aneurysms (AAAs). AAAs represent a potentially life-threatening condition characterized by the localized dilation of the abdominal aorta, often asymptomatic until vessel rupture occurs, which holds a high risk of death [4]. Growing evidence suggests that neutrophil-mediated inflammation, exacerbated by the release of NETs, contributes to the development, progression, and destabilization of AAAs, highlighting the intricate relationship between immune dysregulation and vascular pathology [5].

In this review, we delve into the versatile roles of NETs in CVDs, with a specific focus on their involvement in aortic aneurysms. We begin by elucidating the mechanisms underlying NET formation and function, providing a foundational understanding of their biological features. Then, we explore the role and molecular mechanism of NET formation in various cardiovascular conditions, such as thrombosis, atherosclerosis, myocardial infarction, and COVID-19, before focusing on the specific implications of NETs for the pathogenesis of AAA. Through this exploration, we aim to provide insights into the complex relationship between NETs and cardiovascular pathology, fostering a deeper understanding of the mechanisms driving CVD and AAA development and highlighting potential avenues for clinical translation.

## 2. Methodology

In composing this narrative review, a literature search was conducted using the PubMed database with a comprehensive set of search terms reflected in the titles of the review sections. Retrieved articles were initially screened based on their titles and abstracts to assess their relevance in documenting the role of NETs in CVD pathophysiology, which is the object of this review. Thus, the inclusion criteria encompassed studies investigating the presence and role of NETs in CVDs, in particular thrombosis, atherosclerosis, COVID-19, and myocardial and aortic disease. Notably, reports on NETs in autoimmune conditions (such as vasculitis) were excluded from this review. Priority was given to renowned flagship research as well as recent publications within the last 5 years to ensure the incorporation of the most important and up-to-date research findings. Following the initial screening, selected articles were retrieved in full text and further evaluated for their contribution to the field. Emphasis was placed on including studies with significant clinical or mechanistic insights. Additionally, we considered the diversity of study designs, including experimental studies, clinical trials, and observational studies, to provide a comprehensive overview of the topic and identify potential research gaps.

## 3. Neutrophil Extracellular Traps (NETs)

Brinkmann et al. described NETs in 2004 as extracellular DNA fibers composed of granule and nuclear components that disarm and kill bacteria [2]. Since then, multiple pathways have been proposed as to how exactly these NETs form, mainly controlled through the regulation of intracellular reactive oxygen species (ROS) or calcium. The activation of NADPH oxidase (Nox2), as triggered by the MAPK/ERK signaling cascade [6], leads to the formation of ROS, which promotes the release of myeloperoxidase (MPO) and neutrophil elastase (NE) from the azurophilic granules and their translocation to the nucleus; there they aid in the decondensation of nuclear chromatin and the subsequent rupture of nuclear and cytoplasmic membranes leading to ejection of the NETs into the extracellular space [7]. A rise in intracellular calcium can activate an alternative pathway of NET formation that does not necessarily involve Nox2 [8,9]. The enzyme peptidylarginine deiminase 4 (PADI4) is activated by the calcium trigger to convert arginine residues of histones to citrulline, thereby effectively changing the charge of the residues, which leads to chromatin decondensation and subsequent DNA strand release [10]. In vitro these two pathways can be separately triggered through use of phorbol-12-myristate-13-acetate or by calcium ionophores, respectively [9].

Experiments with phorbol-12-myristate-13-acetate-stimulated neutrophils also revealed that, in addition to Nox2-mediated superoxide production, the induction of autophagy is required for efficient NET formation. Inhibition of either pathway blocked chromatin decondensation and resulted in apoptotic neutrophil death [11,12]. Autophagy was proposed as a mechanism to control the release of NETs through effects on membrane nucleation, ROS production, histone citrullination, and the availability of intracellular substrates [13]. While the majority of the literature documents an essential and supporting role of autophagy in NET formation [14], not all inducers of autophagy can promote NET release [15].

More recently, Chen et al. identified caspase 11 and gasdermin D as central regulators of NET formation: upon neutrophil exposure to cytosolic lipopolysaccharide or gram-negative bacteria, the non-canonical inflammasome signal triggers caspase 11- and gasdermin D-mediated histone degradation as well as permeabilization (pore formation) of the nuclear and plasma membranes, which results in neutrophil rupture and NET expulsion [16].

In addition to chromatin release from the nucleus, which mostly results in neutrophil death, DNA can also be ejected from the mitochondria, e.g., upon cell priming by granulocyte-macrophage colony-stimulating factor and subsequent activation of toll-like receptor 4 (TLR4) with lipopolysaccharide [17]. It is of note that in this form of NET release, the neutrophil does not undergo immediate cell death as the DNA is extruded in vesicles. While mitochondria were observed to translocate to the cell surface for ejection of (mostly oxidized) DNA into the extracellular space [18], mitochondrial ROS was also found to trigger the release of mitochondrial DNA into the cytoplasm, where it activates the cyclic GMP-AMP synthase-stimulator of interferon genes pathway and stimulates NET formation and inflammatory response [19,20].

As their name suggests, NETs are web-like structures that are composed of DNA, histones, and antimicrobial proteins; they can occupy three to five times the volume of condensed chromatin to physically entrap and neutralize pathogens [21]. The DNA serves as a scaffold upon which the other components are assembled, thus providing structural integrity to the NET. Added to that, DNA itself has antimicrobial activity due to its ability to chelate surface bound cations, disrupt membrane integrity, and lyse bacterial cells [22]. In particular, the released histones are known to have cytotoxic properties [23,24]. MPO, NE, and cathepsin G are the primary proteins that originate from the neutrophil granules and are incorporated into NETs for defense against pathogens. MPO helps to create an oxidative environment that aids in the degradation of trapped pathogens through generation of ROS, hypochlorous acid, and nitryl chloride [25]. NE plays a key role in NET release through the cleavage of histones, which leads to chromatin decondensation, as well as through the degradation of microbial proteins [25]. Other proteases such as cathepsin G also aid in the neutralization of microbial components, breakdown of extracellular matrix (ECM) proteins, and recruitment of immune cells to the site of infection/inflammation [26]. NETs may also contain other types of molecules, such as lactoferrin and pentraxins, which additionally modulate the immune response [27,28]. Thus, the local accumulation and coordinated response of various neutrophil granule components contribute to the effectiveness of NETs in host defense [29].

In addition to their direct functions on pathogens, NETs also recruit and activate other immune cells in the modulation of inflammatory processes. Some of the cellular components released are cytokines, chemokines, and danger-associated molecular patterns (DAMPs) [30]. NETs activate macrophages to secrete molecules like interleukin 8 [31], tumor necrosis factor alpha [32,33], and high-mobility group box 1 [34] that act as chemoattractants not only for other neutrophils but also for monocytes, macrophages, and dendritic cells; they can phagocytose pathogens trapped within NETs and process antigenic material for presentation to T cells, thereby initiating adaptive immune responses. The physical structure of NETs provides a scaffold for the adhesion and migration of the various immune cells, facilitating their localization to the site of NET release [35,36]. While NETs may trigger the death of neighboring endothelial cells [37], the NET-bound cytokines and chemokines can also activate these endothelial cells [33], promoting the upregulation of adhesion molecules and the recruitment of circulating immune cells from the bloodstream into the tissue. Collectively, these mechanisms highlight how NETs can shape immune responses in various physiological and pathological settings.

## 4. NETs in Cardiovascular Diseases

Beyond their antimicrobial functions, NETs have emerged as key players in the pathogenesis of various acute and chronic conditions. In the last 10 years, research on the involvement of NETs in CVDs has gained a lot of attention, giving rise to more than 800 original research articles according to PubMed search on 26 February 2024 (excluding reviews and overlapping articles on cancer), with a predominance of thrombosis research and an additional peak of COVID-19 research in 2020–2023. The following sections will summarize the evidence for NETs in distinct cardiovascular conditions, which is further illustrated in Figure 1.

### 4.1. Atherosclerosis

Atherosclerosis is a chronic inflammatory disease characterized by the accumulation of lipid-laden plaques in the arterial wall, and NETs have been found to be present in these atherosclerotic lesions [59]. NETs can promote endothelial dysfunction and vascular inflammation contributing to the initiation and progression of atherosclerotic plaques both by activation and damage of ECs via type I interferon response [60], and through recruitment of other immune cells, mainly macrophages [61].

NETs exacerbate plaque instability by inducing the release and local accumulation of pro-inflammatory cytokines and matrix metalloproteinases (MMPs) [62], which degrade ECM proteins such as collagen, a stabilizing component of the fibrous cap, and therefore lead to increased plaque erosion and the risk of rupture [39]. Among the proteins ejected by neutrophils, histone H4 was found to bind and lyse smooth muscle cells (SMCs), leading to the further destabilization of plaques, attracting more neutrophils in the process, and propagating a chronic inflammatory state [38]; in a mouse model, this detrimental impact could be neutralized by the inhibition of histone H4 interaction with SMCs [38].

MPO accumulating in NETs drives further ROS release and the modification of low-density lipoprotein (LDL) to oxidized LDL, thus promoting the development of foam cells [41]. These lipid modifications and related signaling pathways have emerged as further regulators of NET formation. We and others have shown that oxidized LDL may synergize with various stimuli to enhance NET formation [63,64,65]. Another feedback loop occurs when cholesterol crystals found in atherosclerotic plaques trigger neutrophils to release NETs, resulting in a vicious cycle where NETs prime macrophages for IL-1β production, which leads to the secretion of T-cell-derived IL-17, recruiting more immune cells to the plaque [62]. Recently, NETs were shown to increase lipid accumulation and foam cell formation through the negative regulation of autophagy in macrophages as well as by enhancing NLRP3 inflammasome activity and IL-1β and IL-18 secretion, which further aggravates the inflammatory response and plaque instability [40].

### 4.2. Thrombosis

NETs have emerged as key contributors to thrombosis, the pathological formation of blood clots within blood vessels, by promoting both primary as well as secondary hemostasis. Thus, the mutual stimulation of platelet activation and NET formation was discovered [66]. Furthermore, various components of NETs were found to act as procoagulant molecules, including DNA [43], histones [46,67], and attached granular proteins [66]. These components interact with the coagulation cascade, amplifying the clotting process and promoting thrombus formation. The first publication documenting this link showed that NET markers were detected in the thrombus and in plasma of a baboon model of deep vein thrombosis [46]. NETs were proposed to contribute to thrombus formation and stabilization by the extracellular DNA serving as a scaffold for platelet adhesion and activation, which could be interrupted with DNase 1 or heparin treatment. Additionally, NETs were shown to recruit and bind red blood cells (RBCs) and procoagulant molecules, such as von Willebrand factor, fibrinogen, fibronectin, factor XII, and tissue factor, thereby promoting thrombin-dependent fibrin deposition [46,48,49].

Histones, particularly H3 and H4, contribute to thrombus stabilization by their anti-fibrinolytic effect that reduces plasma clot dissolution by tissue-type plasminogen activator [45,46,47]. When released during NET formation, histones induce platelet aggregation through interactions with TLR4, creating a prothrombotic state [42]. Histones also support thrombin generation by reducing thrombomodulin-dependent activation of protein C, thus downregulating the cleavage of the activated cofactors Va and VIIIa [44].

Neutrophil serine proteases, such as NE and cathepsin G, can similarly enhance platelet activation and coagulation pathways by affecting platelet surface molecules [50] and by degrading inhibitors of coagulation. Upon granule release and NET formation, these serine proteases inhibit multiple anticoagulants, such as tissue factor pathway inhibitor, thereby enhancing the procoagulant activity of tissue factor and factor Xa; neutrophil serine proteases may also directly activate factors XI and XII, initiating the intrinsic pathway of coagulation [51].

As NETs are found in both arterial and venous thrombi, the list of conditions with NET involvement in pathological thrombosis is growing continuously [68]. Recently, NETs have been shown to promote fibrotic thrombus remodeling by enhancing the differentiation of monocytes to activated fibroblasts in chronic thromboembolic pulmonary hypertension [69]. NETs were also identified as major drivers of heparin-induced thrombocytopenia/thrombosis (HIT), where IgG-heparin/platelet factor 4 complexes interact with FcγRIIa on neutrophils to trigger the formation of NET-rich thrombi, which could be blocked with PADI4 inhibitors experimentally [70].

### 4.3. Myocardial Infarction and Ischemic Heart Disease

Considering the role of NETs in thrombosis, their direct or indirect involvement in heart disease was investigated, and NETs were observed in thrombi from patients with acute coronary syndrome [71]. During myocardial infarction (MI), the ischemia-reperfusion (IR) injury leads to an inflammatory response where neutrophils play a crucial role and NETs may contribute to tissue damage and adverse cardiac remodeling [72].

Underlying mechanisms have been revealed for ST-segment elevation myocardial infarction (STEMI). NETs were observed to induce TLR4-mediated monocyte differentiation into CD11b-expressing fibrocytes, which home to the myocardium to contribute to adverse remodeling [55]. In a follow-up study, the granular protein components of NETs were confirmed to be drivers of monocyte recruitment to the culprit site in STEMI patients by triggering monocyte chemoattractant protein 1 release from local endothelial cells [54]. This, in turn, induced more NET formation and monocyte transdifferentiation into fibrocytes for vascular healing and scar formation. Other studies reported that circulating NET markers were elevated in the acute phase of STEMI and high levels of double-stranded DNA were associated with larger myocardial infarcts and adverse left ventricular remodeling, as well as poor clinical outcomes [71,73,74].

Importantly, NETs are an abundant source of pro-inflammatory cytokines and DAMPs, triggering sterile inflammation and exacerbating tissue damage in the ischemic myocardium [75]. In an experimental model of IR, PADI4 knockout mice with impaired NET release were protected from IR injury and had better post-ischemic function in comparison to wildtype mice [75]. In conclusion, NETs may promote plaque destabilization, thrombus formation, and microvascular obstruction in acute coronary syndrome, and they may further exacerbate ischemic injury as well as impair cardiac remodeling post-MI [52,53].

### 4.4. COVID-19

In April 2020, the first studies reported a possible link between NETs and COVID-19, in particular with respect to the severe respiratory complications, thromboembolic events and systemic inflammation associated with the disorder. NETs were found to contribute to the formation of microthrombi in the pulmonary vasculature through platelet-neutrophil interactions, leading to impaired gas exchange and acute respiratory distress syndrome (ARDS) [56]. Plasma MPO-DNA complexes, which are considered as circulating markers of NETs, were elevated in the course of the infection and correlated with disease severity and clinical outcome [56]. Blood samples from patients were further found to have elevated levels of citrullinated histones and cell-free DNA (cfDNA) that correlated with absolute neutrophil count and acute phase markers such as C-reactive protein and D-dimer [57]. Importantly, NETs were revealed to exacerbate lung injury by inducing epithelial and endothelial cell death and by promoting the release of cytokines such as IL-1β and IL-6 [57]. They were thus proposed to contribute to the development of a hyperinflammatory state and of multiorgan dysfunction in severe COVID-19 cases, through lung epithelial cell damage and the release of the pro-inflammatory mediators [58]. High levels of NETs were detected in tracheal aspirates and lung tissue from patients. Neutrophils from patients produced higher levels of NETs than control neutrophils and SARS-CoV-2 was observed to directly induce the release of NETs by healthy neutrophils. This process required angiotensin-converting enzyme 2 and serine protease activity [58].

Recently, several mouse models of COVID-19 were established [76,77]. NETs were highly detectable in the lungs of infected mice, and DNase 1 treatment to dismantle the NETs was shown to improve disease score and to reduce multiorgan injury [76]. Furthermore, signaling via the pro-inflammatory complement component 5a receptor (C5aR1) on neutrophils was found to drive local NET-dependent lung injury in COVID-19 [78]. Both the release of NETs and disease outcome could be controlled with a C5aR1 antagonist in the murine model.

Clinical trials on drug repurposing of recombinant human DNase (dornase alfa, Pulmozyme^®^ nebulizer) to dismantle extracellular DNA/NETs in lung tissue have been initiated and have shown a reduction in DNA-MPO complexes and improvement of respiratory distress in COVID-19 patients [79]. On the other hand, several SARS-CoV-2 vaccines developed to improve immune response, viral clearance and disease severity, were reported to promote rather than mitigate NET formation in this context. A HIT-comparable mechanism leading to the formation of NET-rich thrombi was proposed for vaccine-induced immune thrombotic thrombocytopenia, a condition that arose as a side effect of the SARS-CoV-2 adenoviral vector vaccines ChAdOx1 nCoV-19 (AstraZeneca) and Ad26.COV2.S (Janssen) [80].

## 5. NETs in Aortic Aneurysms

The involvement of NETs in aortopathy, namely thoracic aortic aneurysm (TAA) and abdominal aortic aneurysm (AAA), has been of interest in recent years. Characterized by the abnormal dilation of the aortic wall (affecting intima, media, and adventitia), aortic aneurysms may remain asymptomatic until vessel rupture, which is associated with a high rate of fatal complications [81]. TAA and AAA share certain risk factors such as age, smoking, hypertension, hyperlipidemia, male sex, white race, and a positive family history [82]. Common pathophysiological mechanisms include smooth muscle cell loss or dedifferentiation, ECM degradation, and immune cell infiltration [83]. However, TAA and AAA present in discrete anatomical locations that exhibit different biological properties, primarily due to a distinct developmental origin of the vascular smooth muscle cells [84].

TAA affects the thoracic segment of the aorta, encompassing the aortic root, ascending aorta, aortic arch and/or the descending thoracic aorta, and it is often associated with genetic defects that result in connective tissue disorders [85]. Comparably, AAA is located in the abdominal region of the aorta, primarily the infrarenal segment, where a site of chronic inflammation develops. In contrast to TAA, AAA is frequently accompanied by atherosclerosis and the formation of an intraluminal thrombus (ILT) [81]. The distinct anatomical locations and hemodynamic forces of these aortic segments, in particular the selective ILT formation in AAA, may account for the differing roles of NETs in their pathogenesis. In this section, we will focus on the emerging insights into the involvement of NETs in AAA, given the mounting evidence highlighting their contribution to AAA development as opposed to the limited evidence on their impact in TAA.

Inflammation emerges as a central driver of AAA, orchestrating diverse cellular and molecular responses that contribute to aneurysm formation and expansion. Endothelial dysfunction, often at the site of a pre-existing vascular injury, triggers an inflammatory cascade within the aortic wall [86]. Infiltrating leukocytes, such as neutrophils, macrophages, dendritic cells, natural killer cells, T and B lymphocytes, and mast cells, release cytokines, ROS, and proteases that degrade ECM components and lead to the dedifferentiation or death of SMCs, which weakens the aortic wall [87]. Platelet accumulation and coagulation occur in close proximity due to formation of an ILT, which becomes a further source of entrapped blood cells, secreted proteases, inflammatory cytokines, and fibrinolytic products [88].

The involvement of NETs in AAA has been addressed both in clinical studies and in animal models (see Figure 2 and Table 1). The first report linking NETs to AAA was in 2011, where Delbosc et al. observed that the adventitia and ILT were enriched in NETs, and they found increased levels of cfDNA and MPO-DNA complexes in the plasma of AAA patients in comparison to healthy controls [89]. While this study focused on the potential bacterial trigger of NETs in AAA, other investigations have since shown the presence of NETs in sterile inflammation driving AAA [90,91]. In our analysis of clinical samples, we detected an abundance of NETs in excised aneurysm tissue (wall and thrombus) and found that plasma levels of citrullinated histone H3 (CitH3) were elevated in AAA patients compared to healthy controls and that these normalized after surgical repair [91]. Notably, plasma citH3 held prognostic value for disease progression.

Several mechanisms have been proposed through which NETs may contribute to AAA formation [5]. In a mouse study, NETs were found to be triggered at the early stage of AAA development by macrophage-derived IL-1β which co-localized with NETs in human aortic tissue [90]. Yan et al. [95] observed that NET formation activated plasmacytoid dendritic cells to upregulate the production of type I interferons, which exacerbated inflammation in experimental AAA induction. This was supported in human AAA through detection of neutrophil LL-37 co-localization with plasmacytoid dendritic cells, while healthy aortas were negative for either marker [95]. In addition, NETs were recently proposed to control plasticity and loss of vascular smooth muscle cells. Inhibition of the PI3K/AKT pathway by NETs induced ferroptosis, a type of programmed cell death characterized by high iron-dependent lipid peroxidation, in SMCs and thereby promoted AAA formation [93]. Moreover, NETs were found to induce a synthetic and pro-inflammatory SMC phenotype in a Hippo-YAP pathway-dependent manner, which was associated with AAA development, as reflected in human aneurysm sections [94]. Collectively, these findings highlight the range of mechanisms through which NETs contribute to AAA and suggest NETs as a promising therapeutic target in AAA formation and/or progression.

## 6. Targeting NETs in Cardiovascular Diseases

Approaches to targeting NETs present promising therapeutic avenues for CVD management as they offer potential strategies to mitigate vascular inflammation, thrombosis, and tissue damage associated with various cardiovascular conditions. Several therapeutic interventions have been developed to either prevent NET formation (“upstream NET targeting”) or to degrade and inactivate NET components (“downstream NET targeting”), aiming to attenuate disease progression and improve clinical outcomes in CVD patients (see Figure 3).

### 6.1. Upstream NET Targeting

The inhibition of NET formation through pharmacological agents or biological interventions is one approach to targeting NETs in CVD. The specific knockout of PADI4 in mice, which is known to largely prevent NET formation, showed marked disease abrogation in atherosclerosis [103,104], venous thrombosis [105], heparin-induced thrombocytopenia [106], superficial plaque erosion in ACS [107], myocardial IR [75], and ischemic stroke [108].

Small molecule inhibitors targeting key molecules involved in NET formation, such as NE, Nox2, and PADI4, have shown promising results in preclinical studies. In a mouse model of MI, the pan-PAD inhibitor Cl-amidine abrogated NET formation and reduced arterial thrombosis [98]. Cl-amidine also blocked NET formation and reduced atherosclerosis burden and arterial thrombosis in a murine model [60]. For superficial plaque erosion in atherosclerosis, collagen IV-targeted nanoparticles were developed to deliver the PADI4 inhibitor GSK484 selectively to regions of endothelial cell injury and basement membrane exposure. This approach led to the reduction of NET accumulation at sites of intimal injury and preserved endothelial continuity [99]. Furthermore, a plaque-targeting and neutrophil-hitchhiking liposome (cRGD-SVT-Lipo) was constructed to inhibit NET formation by reducing NE activity with Sivelestat in atherosclerotic plaques, which successfully stabilized lesions and reduced atherosclerosis progression [101]. Inhibition of Nox2 with diphenyleneiodonium chloride (DPI) or GSK2795039 prevented HIT-induced thrombi in an in vivo mouse model [100].

With respect to clinical CVD treatment, upstream NET blockade has not been extensively tested to date. Yet, the SARS-CoV-2 pandemic expedited a number of trials with compounds suited for directly or indirectly reducing NET burden in COVID-19 patients. While no statistical results were reported for the application of the NE inhibitor Alvelestat (ClinicalTrials.gov ID: NCT04539795), blockade of the NE activator dipeptidyl peptidase 1 by Brensocatib did not improve disease status of hospitalized COVID-19 patients [109]. Among established CVD treatments, the beta blocker metoprolol was shown to reduce NET release and ARDS-associated inflammation in critically ill COVID-19 patients [110]. Comparably, STEMI patients undergoing percutaneous coronary intervention treated with adjunct Cangrelor, a P2Y12 receptor blocker to inhibit platelet reactivity, showed a reduction in NETs in peripheral blood and reduced cardiac damage [111]. Regarding the role of gasdermin D in NET formation [112], the clinically available inhibitor disulfiram is being tested in various cancer trials [113,114,115] and is expected to exert NET-dependent and NET-independent effects; however, no CVD clinical trial has utilized this drug to date.

Regarding AAA disease, various approaches to interfering with NET formation were found to be effective in reducing aneurysm development in preclinical models [96]. Use of the pan-PAD inhibitor Cl-amidine significantly attenuated AAA formation in mice [90]. Furthermore, we and others reported that the specific PADI4 inhibitors YW3-56 or GSK484 reduced aorta rupture [92] and prevented aneurysm progression [91,97], which closely matches the clinical setting and therapeutic demand. Importantly, we found that upstream inhibition of NET formation by GSK484 or Nox2ds-tat inhibitory peptide was more effective in controlling AAA disease than downstream interference by dismantling NETs with DNase 1 or by inactivating histone toxicity with HIPe [97]. Our study also established a therapeutic link between NETs and thrombosis in AAA, as NET inhibition was only successful in mice with intramural thrombus formation in aneurysms [97]. Notably, two AAA mouse models were used in our study: angiotensin II (Ang-II) administration to ApoE deficient mice or peri-adventitial application of porcine pancreatic elastase in wildtype mice. While AAA tissue from both models showed the presence of NETs, as marked by CitH3, NET blockade was only effective at attenuating aneurysm progression in the Ang-II model. AAA develops in the elastase model due to an acute local inflammatory insult and then continues into a fibrotic remodeling phase, which is considerably different from the systemically induced, more chronic AAA pathogenesis of the Ang-II model. A recent study highlighted the similarities between established human AAA disease and the Ang-II model through transcriptomic parallels in terms of immune response and metabolic switching; in contrast, the elastase model was classified as a disease initiation model [116]. 

The clinical translation of upstream NET inhibitors for AAA treatment has not been attempted to date. However, a number of trials (including our Vienna MetAAA trial NCT03507413) are currently being conducted worldwide to test the anti-diabetes drug metformin for its efficacy in preventing aneurysm progression. Among the various mechanisms of metformin action, inhibition of NET formation by mitochondrial ROS control has been proposed [117].

### 6.2. Downstream NET Targeting

Therapeutic strategies for NET degradation or clearance were among the first approaches used to control NET-mediated pathology. The use of DNase 1 to cleave extracellular DNA strands is approved for other clinical applications and may therefore be readily implemented in NET-driven diseases. The importance of endogenous DNase 1 and DNase 1-like-3 in NET regulation was first illustrated in knockout mouse models, where intravascular NET accumulation triggered clot formation and obstructed blood vessels in lungs, livers, and kidneys, causing multiorgan damage; this mechanism was then confirmed in patients with severe inflammatory disease and decreased DNase activity [118]. Treatment with DNase 1 was found to reduce the rate of thrombus formation through the reduction of NETs, protecting mice from deep vein thrombosis [49,119]. Administration of DNase 1 to disrupt NETs in atherosclerosis did not alter plaque formation but decreased arterial injury by the processes of plaque erosion and acute thrombosis [107]. Moreover, in chronic thromboembolic pulmonary hypertension, dismantling of NETs with DNase 1 was shown to reduce fibrosis and promote thrombus resolution in a mouse model [69]. Similarly, in a preclinical setting of myocardial IR, treatment with DNase 1 resulted in the improvement of cardiac contractile function [75].

As mentioned above, clinical translation of anti-NET treatment strategies was primarily pursued by aerosolized recombinant human DNase in COVID-19 patients. SARS-CoV-2-infected patients with acute ARDS symptoms showed significantly improved oxygen saturation and recovery for severely ill patients, highlighting the potential therapeutic efficacy of targeting NETs in COVID-19 [79,120,121]. There are several other registered interventional clinical trials with currently no published results on testing the efficacy of aerosolized DNase 1 to dismantle NETs and alleviate respiratory symptoms in COVID-19 (NCT04541979, NCT04445285, NCT05279391, NCT04359654, NCT04409925). DNase is also currently being tested as mono- or adjunct-therapy in the treatment of ischemic stroke (ClinicalTrials.gov ID: NCT05880524, NCT05203224), another vascular disease that is associated with NET release [122].

Similar to DNase 1, heparin, the most commonly used anticoagulant, has been shown in experimental settings to be able to dismantle NETs, resolve NET-platelet aggregates, and destabilize NETs by releasing histones from the NET chromatin [46]. Heparin was further found to neutralize histone-mediated cytotoxicity and reduce sterile inflammation-related mortality in a mouse model of sepsis [123]. Comparably, COVID-19 patients prophylactically treated with the low-molecular weight heparin enoxaparin had reduced blood levels of NET parameters and inflammatory cytokines [102].

A more recently established downstream NET inhibitor is the histone inhibitory peptide HIPe [124]. HIPe functions by binding to the N-terminal tail of histone H4, preventing the interaction of histone H4 with SMCs, which was shown to induce cell membrane lysis. This novel type of cell death is implicated in atherosclerosis pathogenesis, where activated lesional SMCs attract neutrophils and prompt them to release NETs containing histone H4. This results in SMC lysis, inflammation, and plaque destabilization, which was reversed by HIPe administration in a mouse model [38]. Notably, we recently tested whether HIPe would also be protective in AAA disease and found a moderate beneficial effect in mice with an intramural thrombus but no overall reduction of AAA progression [97].

### 6.3. Limitations of Targeting NETs in CVD

Despite the promising therapeutic implications of targeting NETs in CVD, several limitations have so far hindered the wide-ranging translation of NET-targeted approaches into clinical practice. One of the primary concerns is the lack of specificity for several of the above-mentioned compounds for targeting NETs without compromising essential neutrophil functions or other host mechanisms. While experiments based on PADI deficient mice suggested that systemic NET blockade may not necessarily lead to impaired host defense against bacterial or virus infections but may rather limit the response to fungal pathogens [125], less specific anti-NET approaches may indeed compromise immune functions. For example, when MitoTEMPO was applied in a mouse model to target mitochondrial ROS and thereby block NET formation, bacterial burden was increased in the heart and associated with decreased survival [126]. Even the apparently selective approach of PADI4 inhibition may exert off-target effects, leading to unintended adverse reactions or impaired immune responses. For instance, while knockout of PADI4 to inhibit NET formation was found to be protective in multiple disease conditions, as discussed earlier, one study reported that PADI4 deficient neutrophils produced high levels of ROS that led to increased inflammation and tissue damage in the acute phase of MI [127]. Another animal study indicated that NET inhibition by Cl-amidine treatment may on the one hand be protective in myocardial RI but may, on the other hand, be detrimental for post-MI remodeling, indicating that NETs promote cardiac remodeling [128].

Moreover, most of the therapeutic strategies targeting NETs in CVD have been tested in rodent models with regard to disease development, thus raising concerns about the translational applicability of the findings. While clinical trials with recombinant DNase gave promising results in the context of COVID-19 [79,120,121], the majority of NET-targeting clinical studies to date are based on drugs that act pleiotropically and they differ in their outcomes [109]. In a terminated clinical trial (NCT03250689) on chronic obstructive pulmonary disease, the CXCR2 antagonist danirixin was used to inhibit NET release as previously tested through in vitro and in vivo animal experiments [129]. The study was terminated early and the authors reported no difference in sputum NETs between the danirixin and placebo groups [130]. Yet, they suggested that the lack of beneficial effects might be due to a subset of diseased individuals in whom neutrophil activation was CXCR2-independent [130]. This finding highlights the complexity of single-activation-pathway targeting and the challenges in the clinical translation of successful preclinical experiments.

In AAA, most preclinical investigations of NET inhibition focused on disease prevention. When DNase 1 was used to dismantle NETs, it was found that it was successful when given within the first days of AAA induction but failed when administered in an already established disease state [95]. Therefore, it is essential that animal models closely mimic the clinical situation where patients present with formed aneurysms, and effective treatments are required to block disease progression [97,131].

Furthermore, the lack of standardized diagnostic and prognostic markers for NETs in human samples hinders the implementation and monitoring of NET-targeted approaches in clinical practice. While circulating biomarkers, such as cell-free DNA, histones, and neutrophil-derived enzymes, have shown promise in preclinical studies, their clinical utility remains limited by variability in assay sensitivity, specificity, and reproducibility. We recently showed that the widely applied ELISA measurement of MPO-DNA complexes was of poor specificity for NET detection in human plasma [132]. The development of certified, commercially available reagents suited for NET assessment in patient samples would certainly support the translation of anti-NET treatment approaches and companion biomarkers to the clinics.

## 7. Conclusions

This review has elucidated the multifaceted role of NETs in CVD with a particular focus on AAA. We highlighted the mechanisms underlying NET formation and their severe implications in thrombosis, atherosclerosis, myocardial infarction, and COVID-19. We explored the available literature to deduce how NETs contribute to the pathogenesis of AAA by promoting vascular inflammation, thrombosis, and aneurysm progression.

Despite advancements, several gaps remain in our understanding of NETs in AAA and other CVDs, such as how NETs contribute to different stages of disease initiation, progression, and resolution. NETs may exert both protective and detrimental effects depending on the context. Additionally, the impact of genetic factors and of underlying health conditions on the role of NETs in CVD requires further investigation. Achieving selective targeting of NETs while preserving neutrophil function remains a significant challenge in developing NET-directed therapies. The standardization of preclinical models and of NET detection protocols are further steps on the way to the first clinical trials of NET-targeting therapies and biomarkers as reliable tools for clinical decision-making in CVD. Several promising steps towards improved and highly-specific anti-NET drug design have been taken [99,101,133]. The repurposing of already approved drugs that have been found to target NETs, such as DNase 1 [120,121] or disulfiram [134], might expedite clinical translation.

## Figures and Tables

**Figure 1 ijms-25-03983-f001:**
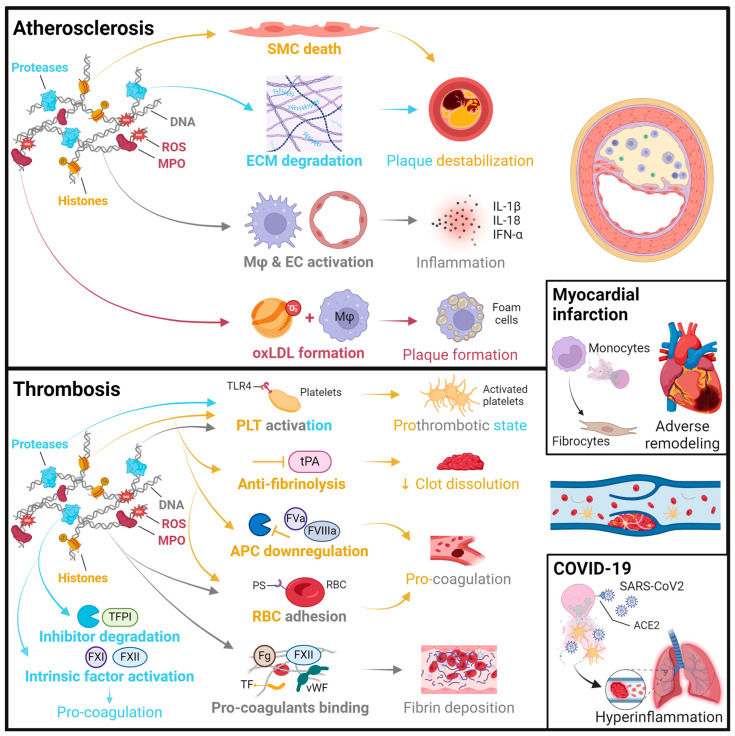
Summary of pathomechanisms of NETs in cardiovascular disease. Atherosclerosis: Histones can induce smooth muscle cell (SMC) death through histone H4 forming pores in the plasma membrane [38]. Proteases, such as neutrophil elastase and cathepsin G, degrade extracellular matrix (ECM) proteins like collagen, leading to plaque erosion or rupture [39]. DNA can activate macrophages (Mφ) and endothelial cells (ECs), leading to the release of inflammatory factors such as IL-1β, IL-18, and IFN-α, which further fosters an adaptive immune response and contributes to plaque progression [40]. Myeloperoxidase (MPO) and reactive oxygen species (ROS) can oxidize low-density lipoprotein (oxLDL), which drives foam cell formation in plaque pathology [41]. Thrombosis: NETs promote thrombosis through their effects on primary and secondary hemostasis. Proteases, histones [42], and DNA [43] activate platelets (PLTs), involving toll-like receptor 4 (TLR4) interaction. Histones and DNA trigger red blood cell (RBC) entrapment and phosphatidylserine (PS) exposure, thus enhancing coagulation. Histones further support coagulation by reducing thrombomodulin-dependent protein C (PC) activation, which decreases the cleavage of the activated cofactors FVa and FVIIIa [44]. They also have an anti-fibrinolytic role through inactivation of tissue-type plasminogen activator (tPA), which impairs clot dissolution [45,46,47]. The DNA serves as a scaffold to bind procoagulant molecules, such as von Willebrand factor (vWF), fibrinogen (Fg), fibronectin, factor FXII, and tissue factor (TF), thereby facilitating fibrin deposition [46,48,49]. Proteases can also promote coagulation through cleaving TF pathway inhibitor (TFPI) [50] and activating factors FXI and FXII [51]. Myocardial infarction (MI): The various pathomechanisms of NETs in atherosclerosis and thrombosis may also contribute to the development of MI [52,53]. Furthermore, granular proteins (NET components) can recruit monocytes to the culprit site by triggering monocyte chemoattractant protein 1 release from local ECs [54]. NETs can also induce TLR4-mediated monocyte-to-fibrocyte differentiation, and these cells can home to the myocardium to contribute to sterile inflammation and adverse remodeling [55]. COVID-19: As described above, NET components may trigger thromboembolic events [56] as well as epithelial and endothelial cell death [57], further contributing to the development of a hyperinflammatory state in COVID-19. Notably, the SARS-CoV-2 virus can directly infect neutrophils in an ACE2 (angiotensin-converting enzyme 2)-dependent manner to promote NET formation [58].

**Figure 2 ijms-25-03983-f002:**
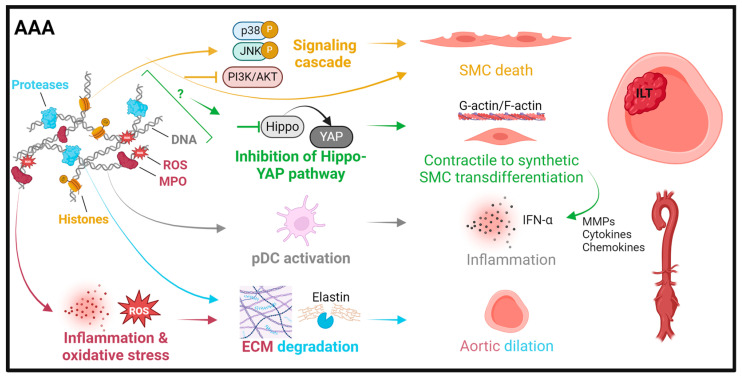
Summary of pathomechanisms of NETs in abdominal aortic aneurysms. Histones can induce smooth muscle cell (SMC) apoptosis through p38/JNK pathway activation [92] or trigger SMC ferroptosis by PI3K/AKT inhibition [93]. NET interference with the Hippo-YAP pathway contributes to SMC transdifferentiation from a contractile to a synthetic phenotype with an associated pro-inflammatory secretion profile [94]. Proteases promote plasmacytoid dendritic cell (pDC) activation and the release of IFN-α, leading to inflammation [95]. Myeloperoxidase (MPO) and reactive oxygen species (ROS) exacerbate inflammation and oxidative stress and further ECM and elastin degradation, which is mainly driven by neutrophil-derived proteases.

**Figure 3 ijms-25-03983-f003:**
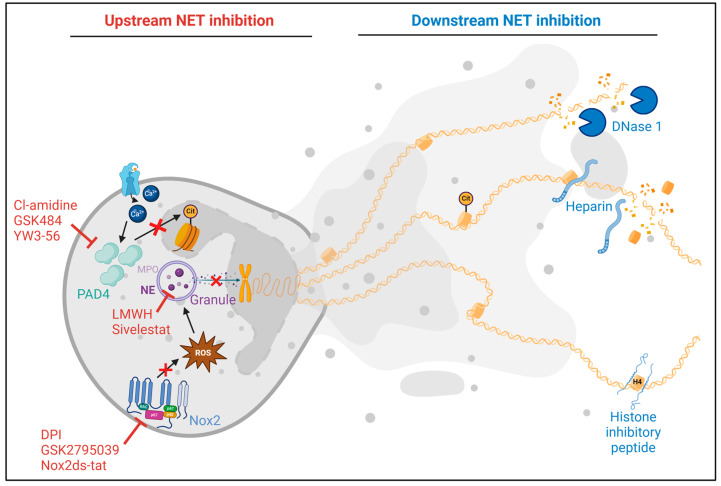
Summary of upstream and downstream inhibition of neutrophil extracellular trap (NET) formation as tested in preclinical models of cardiovascular disease. Upstream inhibitors act by disrupting the NET formation process. These include Cl-amidine [60,98], GSK484 [91,97,99], and YW3-56 [92], which inhibit peptidylarginine deiminase 4 (PADI4), thus preventing histone citrullination and NET release. Inhibitors of the NADPH oxidase 2 (Nox2) pathway, such as diphenyleneiodonium chloride (DPI) [100], GSK2795039 [100], and Nox2ds-tat [97], block ROS generation and subsequent granule release, which is essential for chromatin decondensation. Sivelestat [101] and low-molecular weight heparin (LMWH) [102] target neutrophil elastase (NE) and the release of the granular components. Downstream inhibition focuses on the products of formed NETs. DNase 1 cleaves extracellular DNA, and heparin dismantles NETs and reduces histone-induced platelet aggregation. The histone inhibitory peptide HIPe [38] binds to the N-terminus of histone H4, preventing its lytic interactions with smooth muscle cells.

**Table 1 ijms-25-03983-t001:** Summary of research reports (major findings and employed methods) on NETs in AAA.

Main Findings	Methods of NET Detection	Authors	Year
AAA adventitia and ILT are enriched in NETs. ↑ cfDNA in plasma and tissue-conditioned media. ↑ MPO-DNA complexes in plasma and conditioned media.	Immunodetection of H1, CitH4, and NE in aortic tissue. Picogreen^®^ cfDNA fluorescent measurement. ELISA detection of MPO and MPO-DNA.	Delbosc et al. [89]	2011
NETs contribute to AAA formation in elastase-perfused murine aorta. NET DNA degradation significantly attenuates AAA formation but not AAA progression in mice. ↑ IFN1-driven inflammation through NET-mediated activation of pDCs.	Elastase-triggered AAA in mice: deficiency of dipeptidyl peptidase I or DNase I treatment. Immunodetection of H2B, MPO, DNA in mouse aortic tissue. Immunodetection of neutrophil LL-37 and pDC CD303, CD85g in human aorta.	Yan et al. [95]	2016
IL-1β co-localizes with NETs in human AAA. IL-1β triggers NET release in vitro. NETs are formed in early-stage AAA development in mice with elastase-perfused aorta. IL-1β expression in neutrophils is required for AAA formation in mice. PADI4 inhibition attenuates AAA formation in mice.	Immunodetection of CitH3, NE, DNA and IL-1β in human aortic tissue. Elastase-triggered AAA in mice: IL-1β deficiency and adoptive transfer of WT neutrophils. Immunodetection of CitH3, Ly6B.2, DNA in mouse aortic tissue. Elastase-triggered AAA in mice: Cl-amidine.	Meher et al. [90]	2018
Resolvin D1 reduces aneurysm formation in two mouse models of AAA. ↓ NETs in resolvin D1-treated mice.	Peri-adventitial elastase application to aorta of WT mice or Ang-II treatment of ApoE deficient mice: resolvin D1 treatment. CitH3 Western blot of mouse aortas. Immunodetection of neutrophils, CitH3, NE in mouse aortic tissue.	Spinosa et al. [96]	2018
Plasma levels of CitH3 have diagnostic and prognostic biomarker value for AAA. ↓ CitH3 in plasma after surgical AAA repair. ↑ CitH3 in resected AAA tissue and ILT. Inhibition of PADI4 blocks progression of established AAAs in mice	ELISA detection of MPO, cfDNA-histone complexes, and CitH3 in patient plasma and in tissue-conditioned media. Immunodetection of CitH3, neutrophils (CD66b or Ly6G), DNA in human and mouse aortic tissue. AAA induction by Ang-II in ApoE deficient mice: GSK484 treatment of established AAA disease.	Eilenberg et al. [91]	2021
PADI4 inhibition reduces NET formation, AAA formation and aortic rupture in Ang-II treated mice. NETs trigger SMC apoptosis via p38/JNK pathway.	AAA induction by Ang-II in ApoE deficient mice: YW3-56 treatment. Immunodetection of Ly6G, CitH3, and SM22α in tissue sections and isolated SMCs of mouse aortas: p38/JNK inhibitor application.	Wei et al. [92]	2021
NET release associates with poor clinical outcome: ↑ in ruptured and fast progressing AAA patients. ↑ NETs correlate with ↓ contractile SMCs in mouse AAA tissue. NETs induce synthetic pro-inflammatory SMCs via Hippo-YAP pathway. Loss of PADI4 or YAP alleviates AAA formation in mice.	ELISA detection of CitH3 in human plasma. Fluorometric cfDNA detection in human plasma. Immunodetection of CitH3, MPO, SM22α, and SMA in human and mouse aortic tissue and of CitH3, PADI4, SMA, and YAP in Western blots of human and mouse aortic tissues and isolated SMCs treated with NETs. AAA induction by Ang-II in ApoE deficient mice: PADI4 or YAP gene deletion.	Yang et al. [94]	2023
NET release associates with poor clinical outcome: ↑ in large, ruptured, fast progressing AAA patients. NET inhibition reduces AAA incidence, size and aorta rupture (death) in mice. NET-induced SMC ferroptosis promotes AAA formation. NETs trigger SMC ferroptosis by inhibiting the PI3K/AKT pathway.	Fluorometric cfDNA detection and ELISA of CitH3 and nucleosomes in human plasma/serum. AAA induction by Ang-II in ApoE deficient mice: PADI4 deletion; treatment with DNase I, ferroptosis inhibitor Fer-1, PI3K activator 740 Y-P. Western blots of human and mouse aorta tissue for CitH3, H3, PADI4, and ferroptosis markers like GPX4. Immunodetection of CitH3, MPO, Ly6G in human and mouse aortic tissue sections.	Chen et al. [93]	2023
NET formation is more pronounced at early time points of AAA formation than progression in two mouse models. Upstream NET inhibition is more effective than downstream NET inactivation at controlling AAA progression in mice. NET blockade is more effective in mice that develop an intramural thrombus. Effective NET blockade is associated with reduced vascular remodeling.	Peri-adventitial elastase application to aorta of WT mice or Ang-II treatment of ApoE deficient mice: treatment of established AAA disease with upstream inhibitors (GSK484, Nox2ds-tat) or downstream inhibitors of NETs (HIPe or DNase I). Immunodetection of CitH3, Ly6G, DNA, SMA, and vimentin in mouse aortic tissue. qRT-PCR of AAA tissue for mRNA (ferroptosis, SMC differentiation, proteolysis, inflammation).	Ibrahim et al. [97]	2024

Abbreviations: AAA, abdominal aortic aneurysm; Ang-II, angiotensin II; ApoE, apolipoprotein E; CD, cluster of differentiation; cfDNA, cell-free DNA; CitH3, citrullinated histone H3; CitH4, citrullinated histone H4; DNA, deoxyribonucleic acid; ELISA, enzyme-linked immunosorbent assay; H1, histone H1; H2B, histone H2B; HIPe, histone inhibitory peptide; IFN1, type I interferon; IL-1β, interleukin 1 beta; ILT, intraluminal thrombus; LL-37, cathelicidin; Ly6B.2, lymphocyte antigen 6 complex locus B.2; Ly6G, lymphocyte antigen 6 complex locus G; MPO, myeloperoxidase; NE, neutrophil elastase; NET, neutrophil extracellular trap; qRT-PCR, quantitative reverse transcription polymerase chain reaction; PADI4, peptidylarginine deiminase 4; pDC, plasmacytoid dendritic cell; SM22α, smooth muscle protein 22 alpha; SMA, alpha smooth muscle actin; SMC, smooth muscle cell; WT, wildtype; YAP, yes-associated protein.

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
