# Peer review of "Neutrophil Extracellular Traps in Cardiovascular and Aortic Disease: A Narrative Review on Molecular Mechanisms and Therapeutic Targeting"

_ijms, 2024, doi:10.3390/ijms25073983_

Round 1
Reviewer 1 Report
Comments and Suggestions for Authors
Authors present a review on participation of neutrophil extracellular traps (NETs) in cardiovascular diseases (CVD) with a focus on abdominal aortic aneurism (AAA). The topic is timely and is well presented and referenced in this manuscript. I have rather minor comments as listed.
1. The Figure 1 presents the last 10 years increasing number of original papers about NETs, including CVD. However, this sort of bibliometric statistics do not contribute much more than a preceding sentence mentioning a total and COVID-19 related peak. I suggest a replacement of this figure to a diagram summarizing all the constituents of NETs as described in the Introduction, to help in understanding their role in the inflammation.
2. There are at least 3 well established mouse models of AAA (AngII, elastase or calcium phosphate/chloride injections). Authors could comment on their link to the NETs formation.
3. Although biomarkers of NETs formation are scarce, as correctly mentioned in the Limitations (5.3), would be interesting to learn how is the presence of NETs documented in the aortic wall, especially one affected by aneurysm. Therefore Authors could present a table compiled from references or their own experimental experience on biomarkers, to evidence how contribution of NETs to AAA is studied. Otherwise, the chapter 4 is more about some possible molecular effects of NETs rather than actual presence of them in degenerating aortic wall.
4. Within the Discussion of possible therapeutic intervention, is more reasonable to place Upstream targeting (5.2) before Downstream targeting (5.1).
Reviewer 2 Report
Comments and Suggestions for Authors
The paper provides a comprehensive overview of the role of neutrophil extracellular traps (NETs) in cardiovascular diseases (CVDs), with a particular emphasis on abdominal aortic aneurysms (AAAs). The review covers various aspects of NET formation, function, and their implications in different cardiovascular conditions, such as thrombosis, atherosclerosis, myocardial infarction, and COVID-19. The context is well-organized and provides valuable insights into the complex relationship between NETs and cardiovascular pathology. However, there are a few areas that require clarification and improvement, as detailed below.
§ The processes underlying the formation of NETs are comprehensively detailed, covering both ROS-dependent and ROS-independent pathways. However, it would be beneficial to discuss recent advancements in our understanding of these mechanisms, including any novel regulators or pathways identified.
§ To further improve the comprehensiveness of this section, a concise summary of the regulation of NETosis, focusing on the roles of autophagy and mitochondrial dynamics, would be beneficial. Moreover, it would be advantageous to allocate greater emphasis on the impediments and constraints associated with the therapeutic strategies targeting NETs in cardiovascular diseases. Furthermore, discussing recent clinical trials or experimental studies assessing the anti-NET treatments would offer invaluable insights into the practicality and efficacy of these approaches.
§ Additionally, it would be beneficial to explore the possible consequences of COVID-19 immunization on NET development and cardiovascular wellness, which could elevate the significance of this section.
§ The review presents an extensive discussion on the role of NETs in AAA pathogenesis; however, it falls short in providing experimental validation for certain proposed mechanisms. For instance, the section pertaining to NET-induced SMC phenotype modulation cites mouse model studies but lacks direct experimental evidence in the context of human AAA. To strengthen the manuscript's conclusions, it is recommended to include such experimental validation.
§ Furthermore, while the manuscript briefly acknowledges certain limitations, such as the lack of specificity of targeting compounds and translational challenges, these points could be elaborated upon for greater clarity. Providing specific examples of potential off-target effects or challenges in clinical translation would better contextualize the limitations and enhance the manuscript's overall impact.
Reviewer 3 Report
Comments and Suggestions for Authors
The involvement of neutrophil extracellular traps (NETs) in thrombosis and cardiovascular diseases, including abdominal aortic aneurysm, may open new therapeutic perspectives with important practical implications. For this reason, the theme addressed in the manuscript is particularly interesting.
However, some changes are needed:
- It is necessary to specify from the title that the manuscript is a review. Also, it is absolutely necessary to specify the type of review: systematic or narrative.
- Even if the article does not meet the criteria of a systematic review, certain conditions must be met for a narrative review:
- the object of this research must be clearly specified
- in the methodology (which is completely missing) it is good to specify the databases in which the search was carried out, the searching methods (main and secondary), the time period considered, the number of studies that were identified and those that have were analyzed, the inclusion and exclusion criteria.
3. The analyzed studies must be gathered in a table, including the author, the year of publication and the study result in brief.
For a better understanding of the pathogenic mechanisms, I suggest the authors to use more frequent figures or schemes in the manuscript.
Round 2
Reviewer 2 Report
Comments and Suggestions for Authors
I recommend accepting the manuscript without any further revision.
Reviewer 3 Report
Comments and Suggestions for Authors
The authors took into account the recommendations of the reviewers and made improvements to the manuscript, thus making it suitable for publication.